# Modelling Extended Lactations in Polish Holstein–Friesian Cows

**DOI:** 10.3390/ani11082176

**Published:** 2021-07-22

**Authors:** Agnieszka Otwinowska-Mindur, Ewa Ptak, Joanna Makulska, Olga Jarnecka

**Affiliations:** Department of Genetics, Animal Breeding and Ethology, University of Agriculture in Krakow, al. Mickiewicza 24/28, 30-059 Krakow, Poland; ewa.ptak@urk.edu.pl (E.P.); joanna.makulska@urk.edu.pl (J.M.); olga.jarnecka@urk.edu.pl (O.J.)

**Keywords:** lactation curve, milk traits, Wilmink function

## Abstract

**Simple Summary:**

Mathematical models of lactation curves are functions that describe milk production on each day of lactation. These models are able to predict milk yields as well as provide valuable information applicable in breeding and management decisions. The aim of the present study was to examine different shapes of lactation curves for milk traits (i.e., milk, fat, protein and lactose yields and urea content in milk) modelled by the Wilmink function and by linear or squared functions between 306 and 400 days in milk (DIM). The results suggested that the course of extended lactations could be modelled by a nonlinear model, for example, the Wilmink function, for up to 305 DIM, and the linear or squared function could be more appropriate afterwards.

**Abstract:**

The objectives of this study were (1) to examine different shapes of lactation curves for milk, fat, protein and lactose yields and urea content in milk fitted by the Wilmink function using the test-day (TD) records and (2) to find the function that best describes test-day records beyond 305 days in milk (DIM) for Polish Holstein–Friesian cows. The data were 6,955,768 TD records from the 702,830 first six lactations of 284,193 Polish Holstein–Friesian cows. Cows calved in 19,102 herds between 2001 and 2018. The following functions were fitted to TD data from each lactation: (1) Wilmink model fitted to the whole lactation, (2) Wilmink model fitted to TD records from 5 to 305 DIM and linear function fitted to TD records from 306 to 400 DIM, (3) Wilmink model fitted to TD records from 5 to 305 DIM and squared function fitted to TD records from 306 to 400 DIM. The present study showed that urea content in milk was modelled slightly worse than other milk traits. The results suggested that the course of lactation could be successfully modelled by a nonlinear model, for example, the Wilmink function, for up to 305 DIM, and by the linear or squared function afterwards.

## 1. Introduction

Mathematical models of lactation curves are functions that describe milk production on each day of lactation. These models are able to predict milk yields as well as provide valuable information that is applicable in breeding and management decisions. They could be used for many practical purposes such as health monitoring, individual feeding, genetic evaluation, determining optimum strategies for artificial insemination and replacement of dairy cows [1,2].

Mathematical modelling has been often used in the analyses of test-day data in dairy cattle [2,3,4,5,6,7,8]. The usefulness of this method depends on how well the functions can mimic the biological process of milk production [3], so selection of the lactation curve model is associated with balancing between the fitting properties and the requirements for biological interpretations [9]. Over the past decades, a variety of models have been proposed to describe the standard lactation curve pattern of milk production. Early studies used parametric functions, such as Wood [10], Wilmink [11] or Ali and Schaeffer [12]. Some of these models, i.e., the Wood and Wilmink functions, have a biological interpretation of parameters [2]. Functions such as Legendre orthogonal polynomials and splines have been proposed as possible alternatives to parametric models describing the lactation course [1,13]. Properties of functions applied to model lactation curves were examined, not only in terms of goodness of fit, but also for indicating peak milk yield and a day of peak milk yield as well as for estimating 305-day cumulative milk yields [14,15]. The existence of lactation curves with shapes markedly different from the standard was usually neglected or considered as biologically atypical [2]. In practice, lactation curves may vary in shape. One option is a reversed shape, typical for fat and protein contents in milk, with an initial decreasing phase to a minimum, followed by an increase until the end of lactation. Other options are continuously increasing or continuously decreasing curves with no lactation peak or curves characterized by the occurrence of an additional peak later in the course of lactation. The latter shape may be associated with seasonal effects and may characterize the cows calving in autumn in pasture-based farming systems [1]. The occurrence of different shapes of lactation curves for milk yield of dairy cows has been examined by Macciotta et al. (2005) and Otwinowska-Mindur and Ptak (2016) [2,16]. The former found that the three-parameter models, i.e., Wood and Wilmink functions, were able to detect two main groups of curve shape: standard and atypical. The latter confirmed that those two types of lactation curves (standard and atypical) could be successfully modelled by the Wilmink function.

The functions used for modelling milk yield during lactation can also be applied to illustrate changes of fat and protein yields or other milk constituents across lactation [14,15]. Quinn et al. (2006) proposed the Wilmink function to model fat and protein contents in milk [14]. Otwinowska-Mindur et al. (2014) found that the different models, i.e., Ali and Schaeffer [12], Guo [17], Wilmink [11] functions, and third- and fourth-order Legendre polynomials [18], behaved similarly in describing fat and protein yields during lactation as those used for milk yield [15]. Additionally, those authors concluded that if breeders wanted to obtain more information about the course of a cow’s lactation, they should consider the function that best fits to the data, i.e., fourth-order Legendre polynomials or Ali and Schaeffer’s model. To our knowledge, the literature reports on the analysis of different shapes of lactation curves for fat, protein or lactose yields and urea content in milk are still sparse.

In the last two decades, a tendency for extending the length of lactations of dairy cows beyond 305 days in milk has been observed in many countries [4,7,19,20]. This phenomenon takes place also in the Polish Holstein–Friesian population. Ten years ago, the records between 305 and 350 days in milk (DIM) constituted about 4%, and those between 350 and 400 DIM only less than 2% of all test-day data [8]. During the last decade the situation changed and now about 12% of all TD records is between 305 and 400 DIM. Dematawewa et al. (2007), while comparing nine lactation models, found that all of them satisfactory described milk, fat and protein yields in extended lactations. The authors recommended two simpler functions developed by Rook [21] and Wood [10] and pointed out that the choice of a particular function depended on the potential future use of the fitted lactation curves [7]. On the other hand, according to some reports, the functions used to model standard (305-d) and extended lactations ranked differently [4,8]. VanRaden et al. (2006) observed that lactation models such as Wood [10], Rook [21] and Dijkstra [22] were poorly fitted for production beyond 305 DIM and underestimated the observed yields during the later stages of extended lactations. Of note, the modified Dijkstra [22] function presented convergence problems but gave the asymptotic level to which both milk yield and constituents tended in lactations exceeding 305 DIM [19]. According to Otwinowska-Mindur et al. (2013), the goodness of fit of different lactation curve models was worse beyond 305 DIM due to a small number of data available in that period of lactations [8]. Consequently, they recommended the fourth-order Legendre polynomials [18] for modelling 305 d lactations and the Ali and Schaeffer [12] function to describe extended lactations [8].

We hypothesize that the course of lactation, both for milk yield and milk constituents, can be modelled by nonlinear models until 305 DIM, and the linear or quadratic (squared) functions might be used for modelling lactation shape beyond 305 DIM. Therefore, the objectives of this study were (1) to examine different shapes of whole lactation curves in Polish Holstein–Friesian cows when the Wilmink function was fitted to test-day milk, fat, protein and lactose yields and urea content in milk, and (2) to find the function that best describes test-day records beyond 305 days in milk.

## 2. Materials and Methods

The data were 6,955,768 test-day (TD) records from the 702,830 first six lactations of 284,193 Polish Holstein–Friesian cows registered in the Polish national recording system (SYMLEK). The data were made available by the Polish Federation of Cattle Breeders and Dairy Farmers. Records included five traits, i.e., TD milk, fat, protein and lactose yield, and milk urea content. There were less than 1% TD records in which only milk yield was available. There were 4–16 TD records per lactation per cow in the data file. Table 1 presents the descriptive statistics of the data.

Cows were calved in 19,102 herds in the years from 2001 to 2018. The following restrictions on data were imposed: TD yields between 5 and 400 DIM, lactations lasting at least 200 days, daily milk yields between 1 and 85 kg, fat content from 1.5 to 9.0%, protein content from 1 to 7% and lactose content from 1.00 to 9.99%.

Lactation curves were modelled separately for each cow in each of first six lactations. To fit the functions a nonlinear least squares method with the Levenberg–Marquardt algorithm was used. Lactation curves were modelled in Python [23] with NumPy [24] and SciPy [25]. The following functions were fitted to the TD data from each lactation:1.The Wilmink [11] model (WIL) fitted to the records from the whole lactation
$$\begin{array}{c} y\left(t\right)=a+b\;\cdot \;t+c\;\cdot \;e^{-0.05\;\cdot \;t} \end{array}$$
where:*t*   —day in milk (DIM);a,b,c  —parameters to be fitted;y(t)   —milk, fat, protein or lactose yield, or urea content in milk at DIM *t*.2.Wilmink model (WIL305) fitted to TD records from 5 to 305 DIM in lactation and:(a)Linear function (LIN) fitted to TD records from 306 to 400 DIM in lactation, assuming that at least one TD record beyond 305 DIM occurred. To fit LIN, the last TD record before 305 DIM and all TD records beyond 305 DIM were used;(b)Squared function (SQRT) fitted to TD records from 306 to 400 DIM in lactation. In this case, at least two TD records beyond 305 DIM were required. To fit SQRT, the last TD record before 305 DIM and all TD records beyond 305 DIM were used.

The parameters (*a*, *b*, *c*) of the Wilmink function have biological interpretations related to the shape of the lactation curve: parameter *a* is associated with the level of production, *b* with production decrease after peak yield, and *c* with production increase towards the peak [11]. The course of the lactation curve and its shape depends on the combination of the signs of the two parameters responsible for the increasing (parameter *c*) and decreasing (parameter *b*) phases of lactation. The different shapes of lactation curves were tested based on the information given in Table 2.

The criteria of the goodness of fit used to compare models were as follows:1.Mean Error (ME $=\frac{\sum e_{i}}{n}$);2.Mean Squared Error (MSE $=\frac{\sum e_{i}^{2}}{n}$);3.Mean Absolute Error (MAE $=\frac{\sum |e_{i}|}{n}$);4.Pearson’s correlation (R) between the measured ($y_{i}$) and estimated (${\hat{y}}_{i}$) milk yields;5.Quotient between the error sum of squares and the observed sum of squares (Q $=\frac{\sum e_{i}^{2}}{\sum y_{i}^{2}}$), with lower values indicating closer similarity between the true ($y_{i}$) and estimated (${\hat{y}}_{i}$) values.

The errors ($e_{i}$) were calculated as the differences between the true, i.e., measured ($y_{i}$), and estimated (${\hat{y}}_{i}$) milk yields.

## 3. Results

Examples of lactation curves with standard shape for milk yield, fitted with the use of each of the compared models, i.e., WIL function fitted using records from whole lactation, WIL305 fitted using TD records up to 305 DIM, and LIN or SQRT functions fitted to records beyond 305 days in milk are given in Figure 1.

The values of different criteria of goodness of fit of WIL305 and WIL used to fit lactation curves up to 305 DIM are shown in Table 3. According to all five criteria, i.e., ME, MSE, MAE, R and Q, both models offered similar fit to TD data from standard 305 d lactations. However, the WIL305 model was slightly better than the WIL function, which could be explained by the fact that beyond 305 DIM few TD data were available, so the fit of the WIL function was less accurate in those days. For all analysed traits, ME values were smaller for the WIL305 function than for the WIL function. The values of MSE and MAE were similar for each of the analysed traits. Values of traits estimated by both models were highly correlated with true values (R > 0.9) except urea content in milk (R = 0.8). The values of the Q criterion were the lowest when milk and lactose yields were modelled by the WIL305 function, showing the best compatibility between true and estimated yields for those two traits. The highest Q values (i.e., the worst correspondence between true and estimated values) were found for urea content in milk modelled by both functions with a slightly better fit of WIL305 over WIL.

It was interesting to see how the WIL and WIL305 functions modelled the lactation curve during the first 305 DIM depending on the shape of the curve. Examples of different shapes of first lactation curves modelled using the WIL305 function are given in Figure 2. The Wilmink function, i.e., WIL and WIL305, classified the shapes of individual lactation curves for milk, fat, protein and lactose yield and urea content in milk into two main groups: standard (62–65%) and atypical (about 31%). Reversed lactation curves and continuously increasing curves occurred very rarely. For example, modelling milk yield using the WIL function resulted in only about 2.5% reversed curves and in less than 1% continuously increasing curves. In the case of the WIL305 function, more curves (about 4%) were modelled as reversed and a similar number (about 1%) were modelled as continuously increasing curves. The distance from calving to the first TD could affect the number of atypical curves identified by the WIL and WIL305 models, because the parts of a fitted curve before the first data point and after the last one were extrapolated. The less days was between calving and first test the more lactations had a typical shape. Thus, when the first TD was performed at the very beginning of the lactation, i.e., before 20 DIM, more than 69% and 66% of curves were modelled as standard curves using the WIL and WIL305 models, respectively. When the first TD was between 20 and 50 DIM the standard shape occurred in 62% of curves modelled using the WIL function and about 59% of curves modelled using the WIL305 function. The number of standard shaped lactations was even lower when the first test happened after a peak, i.e., in the declining part of lactation (after 50 DIM). In such a case, the WIL function modelled only about 51% and WIL305 46% of curves with standard shapes. Table 4 presents values of all criteria of goodness of fit for the WIL and WIL305 functions and different shapes of milk yield lactation curves. These values indicate that both functions better predicted milk yields for standard and atypical lactation curves than for two others (reversed and increased curves). For example, the values of correlation (R) for reversed and continuously increasing lactation curves were between 0.91 and 0.93, whereas for standard and atypical curves, between 0.95 and 0.96. Additionally, it can be seen that the 305 d lactation curves modelled by WIL fitted only slightly worse to data.

In the extended lactations, the test-day records beyond 305 days in milk were modelled using three different functions: WIL, LIN and SQRT. Table 5 shows the goodness of fit criteria used for comparing these three models. It is worth mentioning that the WIL function gave the worst fit of each trait. This might be mainly due to few data being collected between 305 and 400 DIM (less than 12% of all TD data), especially during the last 30 days before 400 DIM (less than 3% of all TD records). Another reason could be the fact that yields in days getting closer to 400 DIM were relatively stable. So, we decided to use the linear and squared functions for modelling the lactation curve in this period of lactation. For all analysed traits, the quadratic function (SQRT) seemed to be the best from all three compared functions (WIL, LIN, SQRT) according to five criteria, i.e., ME (absolute value), MSE, MAE, Q and the correlation (R). The advantage of using SQRT for modelling the course of lactation in the period between 305 and 400 DIM was low data requirements: when 2–6 TD yields were available in those days, the SQRT function better fitted the data than the LIN function. It is worth mentioning that, if the milk yield was modelled by the SQRT function, 56% of all models were concave functions, i.e., the coefficients at the second power were negative, whereas the rest of models were convex functions (with positive quadratic coefficients). All those coefficients were small, in range from −0.18 to 0.25; therefore, the arms of fitted parabolas were wide open, causing slow changes in yields in days beyond 305 DIM.

Table 6 shows the comparison of three functions fitted to the TD milk yield beyond 305 DIM, depending on the number of TD records in this part of lactation. As mentioned above, the LIN and SQRT functions fitted better to data in this period than the WIL model. The LIN function could be used even with only one TD record beyond 305 DIM. However, when 2 to 6 TD records were available, the SQRT function modelled this part of lactation slightly better than the LIN function. Similar conclusions could be drawn about other traits, i.e., fat, protein and lactose yields and urea content in milk.

## 4. Discussion

Values of five different criteria were calculated to examine the suitability of compared models, i.e., WIL, WIL305 with LIN or SQRT functions, used to describe the course of lactation for milk, fat, protein and lactose yields as well as urea content in milk (Table 3 and Table 5). The presented results indicated that all models were well-fitted to the TD data; however, urea content in milk was modelled the worst. Lactation curves for milk traits other than daily milk yield have received little attention in the literature. Dematawewa et al. (2007) observed that fat and protein yields in milk could be described by the same function as milk yield [7], which, generally, was in agreement with our results. However, in our study, modelling TD fat and protein yields was slightly poorer than modelling milk and lactose yields. Additionally, Dematawewa et al. (2007) concluded that when selecting a function, one should take into account the relationship between the complexity of calculations affecting the possibility of its practical use and the type of information to be obtained from the course of extended lactation [7].

Assuming that curves for milk and milk constituents are linked since milk is a mixture of fat, protein, lactose and other constituents dissolved or suspended in water, Silvestre et al. (2009) have jointly undertaken a study on the shapes of lactation curves for different milk traits (milk, fat and protein yields as well as fat and protein percentages). They found that the lactation curve could be described as a cluster of five curves fitted by one single model. In 19.3% of lactations, the curves for milk, fat and protein yields followed the standard shape, while for fat and protein contents in milk, they had the reversed standard shape [26]. This could be explained by the positive correlation of milk yield with fat and protein yields and the negative correlation of milk yield with milk fat and protein percentages, found by Miglior et al. (2007) [27]. Some papers examined the relationship between test-day milk production traits and somatic cell count (SCC) in milk. In the study of Cole et al. (2009), several functions were fitted to test-day somatic cell score (SCS) to identify the most suitable model for those data. It was found that persistency of SCS had low correlations with milk, fat, and protein yields across six dairy breeds [28]. Yamazaki et al. (2009) analysed the relationships between the shape of the first-parity lactation curve and udder disease incidence at different stages of lactation by using daily milk and disease records [29]. Results obtained by Green et al. (2004) suggested that using measures of variation and maximum cow SCC in lactation would enhance the accuracy of predicting clinical mastitis [30].

Numerous studies indicated that the models suitable to describe standard 305 d lactations were often not adequate to describe extended lactations [4,7,31]. Although Otwinowska-Mindur et al. (2013) recommended the Ali and Schaeffer function for modelling curves of extended lactations in Polish Holstein–Friesian cows when all TD records from 5 to 400 DIM were available, the part of the lactation beyond 305 DIM proved to be much more poorly modelled than the period until 305 days [8]. Therefore, in this research, we decided to divide lactation into two phases: until 305 DIM and beyond 305 DIM. Considering the relatively “stable” milk yields in the days leading to the end of lactation, we tried to model yields at that period with use of linear or squared function in addition to a typical nonlinear model of lactation curve, i.e., Wilmink function, which was used for modelling the first 305 days in milk.

As expected, the Wilmink function (WIL) used for all TD records available during lactation gave the worse fit than LIN or SQRT functions. According to Druet et al. (2003), the Wilmink model, as a three parametric curve, was not a very flexible function and could give atypical curves when some yields at the end of lactation were particularly high, i.e., outliers [5]. Thus, to obtain more accurate estimates of milk traits at later stages of lactation, the use of WIL305 and LIN or SQRT seemed to be more appropriate. Another reason to use linear or squared functions was that they could be more easily fitted to few data usually collected between 305 and 400 DIM than parametric models. The additional advantage of linear function was the possibility to apply it, even if only one TD yield after 305 DIM was recorded.

The physiological processes of extended lactations seemed to be different from those of standard, i.e., 305 d lactations. A considerable extension of average lactation length was usually associated with an increased milk yield, greater peak and persistency [7,32]. According to Kopec et al. (2021) cows with longer lactations were able to produce almost the same amount of milk in the period between 305 and 732 days of lactation as they produced in the first 305 DIM [31]. Knight (2005) pointed out that the amount of milk produced in two extended lactations surpassed that produced in three standard 305 d lactations if persistency in the extended lactations was improved from 2% decline per week to 1% [33]. It was found that persistency is significantly and negatively correlated with peak milk yield, but could be distinctly improved by frequent milking and an adequate feeding strategy [33,34]. The maximum milk yield and the day when maximum milk yield was achieved during lactation differed depending on breed and farming conditions but usually that day in the extended lactations was slightly later than in the standard lactations [7,13]. Extended lactation resulted from longer days open and/or shorter drying-off periods. Longer days open was caused by difficulties in fertilizing a cow due to post-partum negative energy balance or by the farmer’s decision to extend the voluntary waiting period (VWP). Extension of VWP contributed to the reduction in the incidence of mastitis as well as metabolic and reproductive diseases which were the most numerous in the period from the onset to the peak of lactation [33,35]. Moreover, a longer VWP might prevent drying off in still high daily milk production, which could negatively affect the health status of udders in the current dry period and the subsequent lactation [36]. Rajala-Schultz et al. (2005) found that for every 5 kg increase in milk production at dry off, the risk of intramammary infections was elevated by 77% [37]. Delaying time of artificial insemination and hence managing cows for extended lactations also minimized the negative impact of pregnancy on milk yield [4,7,28].

As can be seen from the above, the replacement of many short lactations with fewer longer ones in high-yielding cows may improve the reproduction efficiency, reduce artificial insemination cost, disease incidence and the total number of parturitions. This, in turn, results in lower exposure to a high-risk period associated with calving and may contribute to higher cow longevity [33,34,38,39]. However, extension of days open and, consequently, extension of lactation are not beneficial in case of less productive multiparous cows because they show adequate reproductive efficiency early in lactation and decreased persistency during lactation [40].

Thus, the knowledge on lactations extended beyond 305 days and ability to model them are indispensable for optimal dairy herd management, especially for making well-informed decisions on cow feeding, health, breeding and replacement [1,2,13,31,32].

As pointed out by many researchers, the choice of the best lactation model was always a kind of compromise between goodness of fit and other function properties, such as flexibility or robustness as well as computational considerations [3,5,13]. The usefulness of mathematical functions also depended on how well they could mimic the biological process of milk production [3], so selection of the model involved balancing between the fitting properties and the requirements for biological interpretation [9]. It should be emphasized that not all models are suitable to describe all shapes of lactation curves. For example, the Wilmink function is not able to detect curves having an additional peak later in the course of lactation. This shape of lactation is typical for cows calving in pasture-based farming systems [1,41,42]. Therefore, modelling extended lactations of pasture-based dairy cows requires models that are able to detect a double peak, such as, e.g., more flexible fourth-order Legendre polynomials [18]. In the present study, fitting with both WIL and WIL305 models gave similar frequencies of four shapes of lactation curves (Table 4). The obtained results (62–65% of lactations with standard shapes) were consistent with the findings of Macciotta et al. (2005) and Otwinowska-Mindur and Ptak (2015) [2,43]. Macciotta et al. (2005) reported that lactation curves with standard shapes represented about 64% of all lactations, also when the Wilmink function was used [2]. Previous studies of Otwinowska-Mindur and Ptak (2015) carried out on Polish Holstein–Friesian population proved the dependency of the shape of lactation trajectory on lactation number. The largest percentage share of standard shape curves (63%) occurred in the first lactations, and a lower frequency of those curves was found in later lactations (47–51%) [43]. Additionally, Cole et al. (2009), while estimating lactation curves for first and later parities in 6 breeds of dairy cattle, found that parameters describing the shapes of the curves varied considerably [28]. Studies of many authors, cited by Bouallegue et al. (2014) and Lee et al. (2020), proved that the shapes of individual lactation curves were affected by numerous factors such as genetic background, calving year, calving season, calving age, parity, service period, calving to first test-day interval, feeding, health status, environmental conditions and herd [44,45]. Græsbøll et al. (2016) reported large differences in the shape of Wood [10] lactation curves among more than 600 Danish Holstein herds randomly selected from the approximately 3000 herds covered by the regular milk recordings. Thus, it may be concluded that it is reasonable to adjust for the effect of each herd when modelling the lactation curves across the population [46]. Variation in model parameters found in the research of Tekerli et al. (2000), Atashi et al. (2009) and Bouallegue et al. (2014), among others, demonstrate that practical usefulness of lactation curves in managing individual herds requires taking into account the environmental effects such as, e.g., interaction of herd year of calving, season of calving and age at calving [44,47,48].

## 5. Conclusions

Three mathematical models were compared for accuracy of predicting milk, fat, protein and lactose yields, as well as urea content in milk based on TD records. Milk and lactose yields were best fitted by the used functions. Urea content in milk was modelled with a lower accuracy than other milk traits.

The results suggested that the course of lactation could be modelled by a nonlinear model—for example, the Wilmink function, up to 305 DIM, and the linear or squared function were more appropriate afterwards.

## Figures and Tables

**Figure 1 animals-11-02176-f001:**
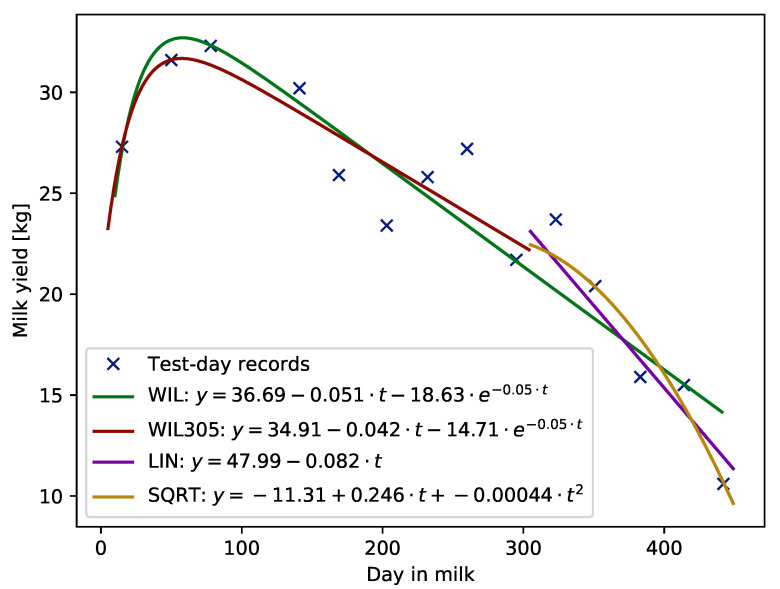
Example of lactation curve with standard shape fitted using Wilmink (WIL and WIL305), linear (LIN) and squared (SQRT) functions.

**Figure 2 animals-11-02176-f002:**
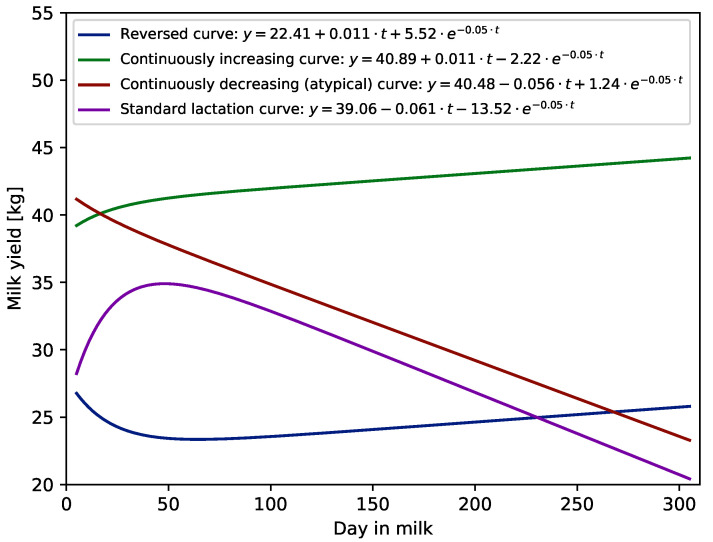
Examples of different first lactation curve shapes fitted using Wilmink (WIL305) function.

**Table 1 animals-11-02176-t001:** Number of test-day (TD) records and lactations and means with standard deviations (SD) for TD milk, fat, protein and lactose yield, and milk urea content, by lactation.

	Number of		Milk (kg)		Fat (kg)		Protein (kg)		Lactose (kg)		Urea (mg/L)	
Lactation	TD Records	Lactations	Mean	SD	Mean	SD	Mean	SD	Mean	SD	Mean	SD
1	1,918,079	247,729	24.24	7.77	0.97	0.30	0.81	0.25	1.19	0.39	219.66	86.08
2	1,253,570	193,902	26.70	10.00	1.08	0.39	0.90	0.31	1.28	0.50	218.87	86.68
3	1,298,760	127,590	27.57	10.45	1.12	0.42	0.92	0.32	1.31	0.52	216.09	86.35
4–6	1,918,079	133,609	26.77	10.26	1.10	0.42	0.89	0.31	1.26	0.51	212.32	85.89
Total	6,955,768	702,830	25.99	9.50	1.05	0.38	0.87	0.29	1.25	0.47	217.43	86.30

**Table 2 animals-11-02176-t002:** Theoretical curve shapes fitted by the Wilmink function.

	Parameter ${}^{a}$	
Curve Shape	b	c
Standard lactation curve	<0	<0
Reversed curve	>0	>0
Continuously increasing curve	>0	<0
Continuously decreasing (atypical) curve	<0	>0

${}^{a}$ Wilmink function: $y\left(t\right)=a+b\;\cdot \;t+c\;\cdot \;e^{-0.05\;\cdot \;t}$.

**Table 3 animals-11-02176-t003:** Goodness of fit of WIL and WIL305 functions to test-day (TD) records until 305 day in milk.

		No. of TD					
Trait	Function	Records	ME ${}^{a}$	MSE ${}^{b}$	MAE ${}^{c}$	R ${}^{d}$	Q *^e^*
Milk (kg)	WIL ${}^{f}$	6,133,988	5.3 × 10${}^{-2}$	8.41	2.06	0.950	1.03
	WIL305 ${}^{g}$	6,133,988	−2.18 × 10${}^{-9}$	7.63	1.95	0.955	0.93
Fat (kg)	WIL	6,104,472	2.90 × 10${}^{-3}$	0.024	0.106	0.913	1.796
	WIL305	6,104,472	−1.92 × 10${}^{-9}$	0.022	0.101	0.919	1.669
Protein (kg)	WIL	6,104,679	3.44 × 10${}^{-3}$	0.010	0.072	0.938	1.140
	WIL305	6,104,679	9.78 × 10${}^{-10}$	0.009	0.068	0.945	1.020
Lactose (kg)	WIL	6,104,668	1.87 × 10${}^{-3}$	0.020	0.102	0.951	1.068
	WIL305	6,104,668	−4.32 × 10${}^{-10}$	0.019	0.096	0.956	0.969
Urea (mg/L)	WIL	6,091,233	0.308	2544.67	36.93	0.813	4.64
	WIL305	6,091,233	−1.84 × 10${}^{-8}$	2339.26	35.40	0.829	4.27

${}^{a}$ ME—mean error, ${}^{b}$ MSE—mean square error, ${}^{c}$ MAE—mean absolute error, ${}^{d}$ R—correlation coefficient between the true and the estimated values, *^e^* Q—quotient between the error sum of squares and the observed sum of squares, ${}^{f}$ WIL—Wilmink function fitted to TD records from 5 to the last DIM in lactation, ${}^{g}$ WIL305—Wilmink function fitted to TD records from 5 to 305 DIM in lactation.

**Table 4 animals-11-02176-t004:** Goodness of fit of WIL and WIL305 functions to test-day (TD) milk yields until 305 day in milk, by different shape of lactation curve.

		No. of						
Function	Curve Shape	Lactations	TD Records	ME ${}^{a}$	MSE ${}^{b}$	MAE ${}^{c}$	R ${}^{d}$	Q *^e^*
WIL ${}^{f}$	Standard lactation curve	459,499	4,039,702	0.0730	8.68	2.10	0.95	0.97
	Continuously decreasing (atypical) curve	221,172	1,908,990	0.0108	7.78	1.98	0.95	1.15
	Reversed curve	17,399	144,244	0.0366	9.12	2.09	0.91	1.61
	Continuously increasing curve	4760	41,052	0.0940	9.45	2.12	0.93	1.27
WIL305 ${}^{g}$	Standard lactation curve	441,583	3,878,838	−1.86 × 10${}^{-9}$	7.85	1.99	0.96	0.88
	Continuously decreasing (atypical) curve	222,451	1,922,683	−4.12 × 10${}^{-9}$	7.02	1.87	0.95	1.01
	Reversed curve	29,743	252,491	8.67 × 10${}^{-9}$	8.72	2.04	0.92	1.42
	Continuously increasing curve	9053	79,976	−5.38 × 10${}^{-9}$	8.56	2.04	0.93	1.04

${}^{a}$ ME—mean error, ${}^{b}$ MSE—mean square error, ${}^{c}$ MAE—mean absolute error, ${}^{d}$ R—correlation coefficient between the true and the estimated values, *^e^* Q—quotient between the error sum of squares and the observed sum of squares, ${}^{f}$ WIL—Wilmink function fitted to TD records from 5 to the last DIM in lactation, ${}^{g}$ WIL305—Wilmink function fitted to TD records from 5 to 305 DIM in lactation.

**Table 5 animals-11-02176-t005:** Goodness of fit of different functions to test-day (TD) records beyond 305 day in milk.

		No. of TD					
Trait	Function	Records	ME ${}^{a}$	MSE ${}^{b}$	MAE ${}^{c}$	R ${}^{d}$	Q *^e^*
Milk (kg)	WIL ${}^{f}$	821,780	−0.40	9.57	2.28	0.89	2.62
	LIN ${}^{g}$	821,780	0.10	3.67	1.23	0.96	1.00
	SQRT ${}^{h}$	673,332	−0.01	1.83	0.75	0.98	0.49
Fat (kg)	WIL	818,006	−0.0217	0.022	0.109	0.86	2.99
	LIN	818,006	0.0046	0.009	0.062	0.94	1.29
	SQRT	669,376	−0.0005	0.005	0.039	0.97	0.66
Protein (kg)	WIL	818,061	−0.0257	0.013	0.086	0.88	2.61
	LIN	818,061	0.0037	0.005	0.047	0.95	1.01
	SQRT	669,427	−0.0005	0.003	0.028	0.98	0.49
Lactose (kg)	WIL	818,061	−0.0140	0.023	0.111	0.89	2.75
	LIN	818,061	0.0050	0.009	0.060	0.96	1.05
	SQRT	669,427	−0.0006	0.004	0.036	0.98	0.50
Urea (mg/L)	WIL	816,018	−2.30	2769.30	40.19	0.78	5.10
	LIN	816,018	−0.24	1417.32	25.33	0.90	2.61
	SQRT	667,270	0.03	792.30	16.49	0.94	1.46

${}^{a}$ ME—mean error, ${}^{b}$ MSE—mean square error, ${}^{c}$ MAE—mean absolute error, ${}^{d}$ R—correlation coefficient between the true and the estimated values, *^e^* Q—quotient between the error sum of squares and the observed sum of squares, ${}^{f}$ WIL—Wilmink function fitted to TD records from 5 to the last DIM in lactation, ${}^{g}$ LIN—Linear function fitted to TD records from 306 to the last DIM in lactation. ${}^{h}$ SQRT—Squared function fitted to TD records from 306 to the last DIM in lactation.

**Table 6 animals-11-02176-t006:** Goodness of fit of WIL, LIN and SQRT functions to test-day (TD) milk yields beyond 305 day in milk by number of TD records per cow beyond 305 DIM.

Function	No. of TD Records Per Cow beyond 305 DIM	No. of TD Records	ME ${}^{a}$	MSE ${}^{b}$	MAE ${}^{c}$	R ${}^{d}$	Q *^e^*
WIL *^e^*	1	148,448	−1.279	10.991	2.462	0.887	3.374
	2	226,912	−0.540	9.402	2.254	0.889	2.636
	3	240,288	−0.138	9.122	2.231	0.891	2.426
	4	172,264	0.087	9.099	2.221	0.891	2.361
	5	33,670	0.171	9.945	2.322	0.891	2.451
	6	198	−0.056	7.730	2.096	0.914	2.226
LIN ${}^{f}$	1	148,448	0.000	0.000	0.000	1.000	0.000
	2	226,912	0.100	3.426	1.281	0.960	0.961
	3	240,288	0.133	4.573	1.548	0.947	1.216
	4	172,264	0.126	5.343	1.681	0.938	1.387
	5	33,670	0.133	6.367	1.838	0.932	1.569
	6	198	0.069	4.291	1.612	0.953	1.236
SQRT ${}^{g}$	2	226,912	0.000	0.000	0.000	1.000	0.000
	3	240,288	−0.009	2.224	1.010	0.974	0.591
	4	172,264	−0.023	3.250	1.257	0.963	0.843
	5	33,670	−0.033	4.155	1.451	0.956	1.024
	6	198	0.018	3.141	1.368	0.966	0.905

${}^{a}$ ME—mean error, ${}^{b}$ MSE—mean square error, ${}^{c}$ MAE—mean absolute error, ${}^{d}$ R—correlation coefficient between the true and the estimated values, *^e^* Q—quotient between the error sum of squares and the observed sum of squares, ${}^{f}$ WIL—Wilmink function fitted to TD records from 5 to the last DIM in lactation, ${}^{g}$ LIN—Linear function fitted to TD records from 306 to the last DIM in lactation. ${}^{h}$ SQRT—Squared function fitted to TD records from 306 to the last DIM in lactation.

## Data Availability

Restrictions apply to the availability of these data. Data was obtained from the Polish Federation of Cattle Breeders and Dairy Farmers.

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
