# Peer review of "Modelling Extended Lactations in Polish Holstein–Friesian Cows"

_animals, 2021, doi:10.3390/ani11082176_

Round 1
Reviewer 1 Report
Additional explanaitions of the statistical models used is (in my opinion) needed for the reader to properly understand what was done (I am unclear)

Author Response
Thank you for reviewing the manuscript and we greatly appreciate your suggestions and many valuable comments which helped us to correct the manuscript. The revision was made using “Track Changes” function (trackchanges packages) in LaTeX and are available as violet and underlying text. Thank you also for the linguistic corrections; all of them were included in the manuscript.
Comments:
It would be sensible to explain whether, or not, Wilmink and post 305-d were constrained to interest at 305-d
Response:
In Poland, 305-d lactation yields are still routinely calculated from test-day yields using a Test-Interval-Method (TIM) and made available to the breeders. When we checked the distribution of all test-day (TD) records available for all lactations and having in mind 305-d lactation yields in routine evaluation, we decided to choose 305-d as a “natural” dividing point.
Comments:
Just a suggestion: a Test-Interval-Method (simple straight line between adjacent tests post 305d could have been used, which has the advantage of being easily made to intersect with the Wilmink 305d estimate.
Response:
Thank you for this suggestion. We continue to do research on finding the best method to calculate yields up to 305-d and post 305-d and we are also planning to use TIM method for extended lactations.
Comments:
It would be helpful to the reader to have the model more clearly explained. For example, were the test-days for each cow-parity used to fit a Wilmink curve separately for each cow (or equivalently, interactions between a, b and c, and Cow-parity), or was a random regression model used? I am hoping that the model was not simply a straight fixed effects regression model ignoring the random cow effect(s), but the model as shown and specified is quite unclear, so I do not know.
Response:
The information was added to the Material and Method section, see lines 103-105 and 112-117.
Comments:
It would be useful to indicate how many of the squared curves had a -ve quadratic coefficient, and how many a positive coefficient (the example given in Figure 1 has a -ve coefficient), whilst it might well be that a substantial number of cows might have +ve coefficients for the quadratic,
Response:
The information was added to the Results section (see lines 190-194).
Reviewer 2 Report
The manuscript entitled “Modelling extended lactations in Polish Holstein-Friesian cows” has been carefully evaluated. In this study the authors provide data about mathematical models concerning lactation curves for milk, fat, protein and lactose yields and urea content in milk fitted by the Wilmink function using the test-day records of Polish Holstein-Friesian cows.
Some minor revisions are needed. Please note my comments below.
L27: Please add “artificial” in front of insemination, and use throughout the whole manuscript to stay unique.
L32: Please add “past” in front of decades.
L41: Please add a comma behind practice.
L61-63: The sentence appears wordy, please consider to rewrite.
L69: Please replace the “goodness of fit” with another phrase, i.e. “accuracy of curve alignment” similar, and use throughout the whole manuscript to stay unique.
In the conclusion the authors mention the phrase “slightly worse”, could that be replaced with a more scientific phrase ?
General question, could the fitting of individual curves be calculated to show the significances between the single fitting results ?
Author Response
Thank you for reviewing our manuscript and we greatly appreciate your suggestions and many valuable comments which helped us to correct the manuscript. The revision was made using “Track Changes” function (trackchanges packages) in LaTeX and are available as violet and underlying text.
Comments:
L27: Please add “artificial” in front of insemination, and use throughout the whole manuscript to stay unique.
Response:
It was changed, see lines 27, 272, 275.
Comments:
L32: Please add “past” in front of decades.
Response:
It was changed, see line 32.
Comments:
L41: Please add a comma behind practice.
Response:
It was changed, see line 41.
Comments:
L61-63: The sentence appears wordy, please consider to rewrite.
Response:
The sentence was rearranged, see lines 61-63.
Comments:
L69: Please replace the “goodness of fit” with another phrase, i.e. “accuracy of curve alignment” similar, and use throughout the whole manuscript to stay unique.
Response:
The phrase “goodness of fit” is widely used in the literature so we decided not to change it. See:
- N. Melzer, S. Trißl, and G. Nürnberg. 2017. Short communication: Estimating lactation curves for highly inhomogeneous milk yield data of an F2 population (Charolais × German Holstein). J. Dairy Sci. 100:9136–9142
- S. A. Adediran , D. A. Ratkowsky , D. J. Donaghy , A. E. O. Malau-Aduli. 2012. Comparative evaluation of a new lactation curve model for pasture-based Holstein-Friesian dairy cows. J. Dairy Sci. 95 :5344–5356
- J. Bohmanova, F. Miglior, J. Jamrozik, I. Misztal, P. G. Sullivan. 2008. Comparison of Random Regression Models with Legendre Polynomials and Linear Splines for Production Traits and Somatic Cell Score of Canadian Holstein Cows. J. Dairy Sci. 91:3627–3638
- B. Vargas, W. J. Koops, M. Herrero, J.A.M. Van Arendonk. 2000. Modeling Extended Lactations of Dairy Cows. J Dairy Sci 83:1371–1380
- N. P. P. Macciotta, D. Vicario, A. Cappio-Borlino. 2005. Detection of Different Shapes of Lactation Curve for Milk Yield in Dairy Cattle by Empirical Mathematical Models. J. Dairy Sci. 88:1178–1191
Comments:
In the conclusion the authors mention the phrase “slightly worse”, could that be replaced with a more scientific phrase ?
Response:
The sentence was rearranged, see lines 319-321.
Comments:
General question, could the fitting of individual curves be calculated to show the significances between the single fitting results ?
Response:
Lactation curves in this paper were modelled separately for each cow in each individual lactation (the information is added at line 104). The errors were then calculated as the differences between the real and estimated yields. The criteria of goodness of fit were calculated using these errors.
Reviewer 3 Report
The idea of using values from two different models to model extended lactation curves is new and very interesting and suggestive.
Furthermore, to my knowledge, data on the temporal evolution of the urea content in extended lactations have never been published.
I therefore, consider the paper worthy of attention and of great interest to the readers of the journal.
However, before being suitable for publication, it should be improved in many parts, as some essential elements are incomplete or missing.
In the introduction section, the modelling of the lactation curve is extensively discussed, but the problem to be addressed in the paper is not adequately formalized and most importantly, an adequate review of the models proposed and the results obtained in modelling extended lactations is missing.
With regard to the data used, the dataset consists of a very high number of lactations, certainly sufficient for a robust study.
Considering the large amount of data, I think a stricter editing can be done (why insert TD with 1 kg for milk yield ? .....).
Considering that one of the objectives of the work is to describe the different shapes of the lactation curves, I think that the editing should also cover the distance from calving to the first useful TD.
In fact, this distance is highly correlated to the number of atypical curves identified by the different models.
With regard to the model, several models have been tested and found to be valid in modelling time evolution for both milk yield and milk constituents in extended lactations.
I believe that a new model needs to be compared with models that have proven to work well and its usefulness can only be assessed by the results of this comparison.
in addition, there are some elements of variability (in particular year, herds and parity) that can make the results unrealistic in breeding practice, these elements should be addressed at least in the discussion.
Not all models perform well in all situations, e.g. extending lactation in pasture-based dairy cows needs models that are able to detect a double peak (see for example Kolver et al., 2007; Auldist et al., 2007).
Not all models are adopted because they fit better, e.g. the modified Dijkstra function proposed by VanRaden, presented convergence problems but gives the asymptotic level to which both milk yield and constituents tend for lactations exceeding 305 DIM. This parameter is useful in both selection and management of animal culling (see for example VanRaden et al., 2006; Steri et al., 2012).
These are the minimum elements that should be addressed in a paper presenting a new approach to extensive lactation modelling.
In addition, the presentation of graphs with examples of typical and atypical individual curves for each analyzed phenotype (perhaps as additional material), will make the work more understandable even to readers less experienced in modeling the lactation curve.
Finally, with regard to the presentation of results, I recommend choosing fitting parameters that make a comparison with the relevant literature.
Furthermore, if regressions are made on individual curves, it would be more interesting to also see the fit statistics by fit classes, as averages in this case mean not give a useful indication.
bibliography cited not considered in the manuscript
Haile-Mariam, M., Goddard, M. 2008.
Genetic and phenotypic parameters of lactations longer than 305 days (extended lactations).
Animal, 2:3 325-335.
ES Kolver, JR Roche, CR Burke, JK Kay, PW Aspin.
Extending lactation in pasture-based dairy cows: I. Genotype and diet effect on milk and reproduction.
Journal of Dairy Science, 90 (2007), pp. 5518-5530
M.J. Auldist, G. O’Brien, D. Cole, K.L. Macmillan, C. Grainger.
Effects of varying lactation length on milk production capacity of cows in pasture-based dairying systems.
Dairy Sci., 90 (2007), pp. 3234-3241
VanRaden, P.M., Dematawewa, C.M.B., Pearson, R.E., Tooker, M.E. 2006.
Productive Life Including All Lactations and Longer Lactations with Diminishing Credits.
- Dairy Sci., 89: 3213-3220.
Author Response
Thank you for reviewing our manuscript and we greatly appreciate yours suggestions and many valuable comments which helped us to correct the manuscript. The revision was made using “Track Changes” function (trackchanges packages) in LaTeX and are available as violet and underlying text.
Comments:
In the introduction section, the modelling of the lactation curve is extensively discussed, but the problem to be addressed in the paper is not adequately formalized and most importantly, an adequate review of the models proposed and the results obtained in modelling extended lactations is missing.
Response:
The information about extended lactations was added to Introduction section (see lines 69-84).
Comments:
With regard to the data used, the dataset consists of a very high number of lactations, certainly sufficient for a robust study.
Considering the large amount of data, I think a stricter editing can be done (why insert TD with 1 kg for milk yield ? .....).
Response:
The objective of our study, i.e. finding the function that best describes TD records beyond 305 days in milk is associated with an attempt to replace a Test Interval Method (TIM) with a new method which perhaps will better reflect the shape of extended lactations. The restrictions presented in our manuscript are consistent with those used routinely in Polish HF population so we decided to use all data without any additional restrictions imposed.
Comments:
Considering that one of the objectives of the work is to describe the different shapes of the lactation curves, I think that the editing should also cover the distance from calving to the first useful TD.
In fact, this distance is highly correlated to the number of atypical curves identified by the different models.
Response:
The information about the effect of distance between calving and first test-day on lactation curve shapes was added to the Results section (see lines 161-171).
Comments:
With regard to the model, several models have been tested and found to be valid in modelling time evolution for both milk yield and milk constituents in extended lactations.
I believe that a new model needs to be compared with models that have proven to work well and its usefulness can only be assessed by the results of this comparison.
in addition, there are some elements of variability (in particular year, herds and parity) that can make the results unrealistic in breeding practice, these elements should be addressed at least in the discussion.
Response:
A paragraph was added at the end of the Discussion section (see lines 305-316).
Comments:
Not all models perform well in all situations, e.g. extending lactation in pasture-based dairy cows needs models that are able to detect a double peak (see for example Kolver et al., 2007; Auldist et al., 2007).
Response:
The information was added to Discussion section (see lines 289-294).
Comments:
Not all models are adopted because they fit better, e.g. the modified Dijkstra function proposed by VanRaden, presented convergence problems but gives the asymptotic level to which both milk yield and constituents tend for lactations exceeding 305 DIM. This parameter is useful in both selection and management of animal culling (see for example VanRaden et al., 2006; Steri et al., 2012).
Response:
The information was added to Introduction section (see lines 75-80).
Comments:
In addition, the presentation of graphs with examples of typical and atypical individual curves for each analyzed phenotype (perhaps as additional material), will make the work more understandable even to readers less experienced in modeling the lactation curve.
Response:
The graphs with different lactation curve shapes was added to Results section (see page 7).
Comments:
Finally, with regard to the presentation of results, I recommend choosing fitting parameters that make a comparison with the relevant literature.
Response:
There is a place to present all parameters in Tables 3-6 so if you don’t mind we prefer to leave them all. The fitting parameters, i.e. MS, MAE, MSE and R were used by other researchers in the literature cited here so we had a chance to compare our results with the literature. See:
- A. M. Silvestre, F. Petim-Batista, and J. Colaco: The Accuracy of Seven Mathematical Functions in Modeling Dairy Cattle Lactation Curves Based on Test-Day Records From Varying Sample Schemes. 2006. J. Dairy Sci. 89:1813–1821.
- Dematawewa C.M.B., Pearson R.E., VanRaden P.M., 2007. Modeling extended lactations of Holsteins. J. Dairy Sci. 90, 3924–3936
- Druet T., Jaffrezic F., Boichard D., Ducrocq V., 2003. Modeling lactation curves and estimation of genetic parameters for first lactation test-day records of French Holstein cows. J. Dairy Sci. 86, 2480–2490
- Tekerli M., Akinci Z., Dogan I., Akcan A., 2000. Factors affecting the shape of lactation curves of Holstein cow from the Balikesir province of Turkey. J. Dairy Sci. 83, 1381–1386
The Q parameter was used according to Silvestre et al. (2006).
Comments:
Furthermore, if regressions are made on individual curves, it would be more interesting to also see the fit statistics by fit classes, as averages in this case mean not give a useful indication.
Response:
Lactation curves were modelled separately for each cow in each of lactations (the information is added at line 104). Then the errors were calculated as the differences between the observed and estimated yields. The goodness of fit parameters were calculated using these errors. We hope we understood your question well and our answer is correct?
Round 2
Reviewer 1 Report
The revised manuscript is acceptable to me
Reviewer 3 Report
no other comment